# Dietary Patterns, Serum BDNF and Fatty Acid Profiles in Physically Active Male Young Adults: A Cluster Analysis Study

**DOI:** 10.3390/nu16244326

**Published:** 2024-12-15

**Authors:** Monika Johne, Ewelina Maculewicz, Andrzej Mastalerz, Małgorzata Białek, Wiktoria Wojtak, Bartosz Osuch, Małgorzata Majewska, Marian Czauderna, Agnieszka Białek

**Affiliations:** 1Faculty of Physical Education, Jozef Pilsudski University of Physical Education in Warsaw, 00-968 Warsaw, Poland; monika.johne@awf.edu.pl (M.J.); ewelina.maculewicz@awf.edu.pl (E.M.); andrzej.mastalerz@awf.edu.pl (A.M.); 2Department of Laboratory Diagnostics, Military Institute of Aviation Medicine, 01-755 Warsaw, Poland; 3The Kielanowski Institute of Animal Physiology and Nutrition, Polish Academy of Sciences, 05-110 Jabłonna, Poland; m.bialek@ifzz.pl (M.B.); w.wojtak@ifzz.pl (W.W.); b.osuch@ifzz.pl (B.O.); m.majewska@ifzz.pl (M.M.); m.czauderna@ifzz.pl (M.C.); 4School of Health and Medical Sciences, University of Economics and Human Sciences in Warsaw, 01-043 Warsaw, Poland

**Keywords:** BDNF, brain-derived neurotrophic factor, dietary patterns, physical activity, young adult

## Abstract

**Background/Objectives:** Although physical activity and balanced diet may increase peripheral brain-derived neurotrophic factor (BDNF) concentration, little is known about whether these factors modify BDNF content in physically active individuals and whether the serum fatty acid (FA) profile is related. This study aimed to evaluate quality of diet, identify specific dietary patterns and assess their influence on BDNF and FA levels in serum. It is hypothesized that there is a correlation between diet quality and the concentrations of BDNF and FA in the serum of physically active male individuals. Methods: Physically active young adult male students at Jozef Pilsudski University of Physical Education in Warsaw (Poland) were enrolled. Dietary patterns were identified with cluster analysis and linear discriminant analysis (LDA) based on responses to a validated food frequency questionnaire, KomPAN^®^ version 1.1. **Results**: Consumption of beverages, vegetables, milk, wholemeal bread/rolls, fruit and vegetable juices, butter, tinned vegetables and fruits were significant in the LDA model, in which three clusters were distinguished. Cluster 1 was characterized by more frequent consumption of wholemeal bread/rolls, milk, fruits, vegetables, fruit and vegetable juices and sweetened hot beverages and by significantly greater values for the pro-healthy diet index (*p* < 0.0001) and diet quality index (*p* < 0.0001) compared to Clusters 2 and 3. The diet of Cluster 2 was of the worst quality, as indicated by the higher values of the not-healthy diet index. Cluster 1 had the tendency for the highest BDNF levels (of the best quality of diet), and a tendency for decreased BDNF concentration with an increased physical activity level was observed. **Conclusions**: Physical activity, diet quality and BDNF level depend, correlate and interact with each other to provide both optimal physical and mental health.

## 1. Introduction

The World Health Organization (WHO) claims that physical activity has a positive impact on health. Moderate physical activity (lasting at least 150 min per week), intensive physical activity (lasting at least 75 min per week) or a combination of those two is enough to fulfil WHO recommendations [1]. However, in most societies, people are currently less active than in previous generations, which has a multidirectional impact on overall health [2]. Taking into account positive effects of physical activity, it enhances good health of the cardiovascular and musculoskeletal system, reduces depressive symptoms and promotes better cognitive performance. It is one of the factors that induces brain neuroplasticity through various mechanisms, including increased expression, secretion and transmission of signals via neurotrophic factors, a reduction in stress levels, inhibition of inflammatory processes and improvement in metabolic and cardiac parameters. Brain-derived neurotrophic factor (BDNF) is one of the neurotrophic factors which affects brain plasticity by weakening the degeneration of inducing neurons and influencing neuronal plasticity [3,4]. It belongs to a family of secretory proteins that includes neurotrophins 3 and 4 and nerve growth factor (NGF) and exerts its antioxidant function by activating the tropomyosin-related kinase B (TrkB) receptor [5]. The BDNF-encoding gene is located on the short arm of chromosome 11 (region p13–14), covers ~70 kb and has a complex structure consisting of 11 exons, of which only one, exon 9, contains the protein-coding sequence. The human BDNF gene contains nine promoters and is under the control of transcription factors. BDNF is synthesized in the endoplasmic reticulum as a precursor (prepro-BDNF; ~27 kDa), which, after cleavage of the signaling protein, produces pro-BDNF and then mature BDNF (mBDNF), which can cross the blood–brain barrier. Both variants (pro-BDNF and mBDNF) are biologically active but functionally different [6,7]. The main source of BDNF in mammals is considered the hippocampus, but BDNF can also be produced by different tissues, such as muscle, thymus, heart, liver, vascular smooth muscle cells, lung and spleen [6,8]. BDNF is best known for its influence on the nervous system, as it affects the growth of neurons, contributes to the formation of new synapses and cognitive functions, promotes neuronal survival and differentiation [9,10], plays an important role in the neurobiological development of the brain and regulates synaptic transmission and plasticity in many areas of the central nervous system [11,12]. Additionally, BDNF improves cardiovascular and metabolic functions and is involved in preventing or delaying the development of neurodegenerative diseases [13].

Numerous factors, including age, gender, body weight, use of stimulants (such as tobacco and alcohol), type of physical activity and diet, may influence the level of BDNF in the blood [14,15,16,17,18]. A reduced concentration of BDNF was observed in the blood of patients suffering from mental diseases, such as depression, schizophrenia or dementia [19,20,21], as well as obesity [22], type 2 diabetes [23] or coronary artery disease [5]. BDNF plays a crucial role in controlling body weight and energy homeostasis. It was revealed that increased levels of BDNF reduce food consumption and maintain energy balance [24,25].

One natural way to increase peripheral BDNF concentration is through a balanced diet and physical activity [5,26,27]. The baseline serum level of BDNF was significantly lower in an athlete group compared to a control group with a sedentary lifestyle [28,29]. However, in both groups of individuals, higher BDNF concentrations in the blood were observed after physical exercise [7,28,29,30,31,32,33,34]. A meta-analysis by Szuhany et al. provided reliable evidence that both acute and regular exercise have a significant impact on BDNF levels [35]. Current research on humans also confirms that physical exercise improves cognition and mood and can become an effective method for treating mental and neurobiological diseases [36], with increasing BDNF levels being one of the proposed mechanisms behind this effect [35,37]. Due to its ability to cross the blood-brain barrier, peripheral BDNF exerts a central effect on neuron growth and survival, improves memory performance and reduces the risk of neurodegenerative diseases [35,38,39].

Diet can also promote or inhibit neuronal plasticity. It plays a key role in the treatment of mental disorders because it has been shown that the lipid composition of the cell membranes of nervous system cells, especially the fatty acid (FA) profile and content, greatly influences a person’s mood. The central nervous system (CNS) is characterized by the highest content of FA, immediately after adipose tissue [40]. Numerous studies have demonstrated a connection between the consumption of *n*-3 polyunsaturated fatty acids (*n*-3 PUFA) in the diet and the occurrence of mental disorders [41,42,43]. Three *n*-3 PUFA, eicosapentaenoic acid (c5c8c11c14c17 C20:5, EPA), docosahexaenoic acid (c4c7c10c13c16c19 C22:6, DHA) and α-linolenic acid (c9c12c15 C18:3, ALA) have also been proven to improve cognitive function and mood through increased neurogenesis and synaptic protein expression [44]. Furthermore, a diet rich in DHA has been found to increase BDNF levels in the hippocampus, thereby enhancing cognitive function in rodent models of brain injury [45]. DHA improves cognitive abilities by accelerating synaptic plasticity and modifying the fluidity of the synaptic membrane [46]. Thus, a diet rich in polyphenols, which exert anti-inflammatory and antioxidant effects, works synergistically with *n*-3 FAs to increase BDNF expression, neuronal plasticity and survival [21,27]. However, the influence of diet and dietary factors on BDNF has not been investigated in relation to physical activity levels, and little is known about how dietary factors modify BDNF content in physically active individuals and whether the FA profile of serum corresponds with BDNF levels. Since proper nutrition is one of the most important factors influencing sport performance and is commonly used to improve sport results, it is essential to explore how diet influences BDNF and FA levels in physically active individuals. Therefore, the main objective of this study was to evaluate the diet quality of physically active young adults (by identifying specific dietary patterns and significant dietary factors responsible for distinguishing them) and to assess their influence on BDNF and FA levels in serum. Additionally, the study aimed to analyze any potential relationships among the examined parameters. It was hypothesized that there is a correlation between diet quality and the concentrations of BDNF and FA in the serum of physically active male individuals.

## 2. Materials and Methods

### 2.1. Study Design and Characteristics of Participants

The whole study was approved by the Bioethics Committee at the District Medical Chamber in Gdansk (resolution no. KB-2/21) for studies involving humans and was performed in accordance with the ethical standards laid down in the 1964 Declaration of Helsinki and its later amendments. The study was conducted in the Faculty of Physical Education, Jozef Pilsudski University of Physical Education in Warsaw (Poland). Physically active young adults (men, *n* = 182) were included in the study. Inclusion criteria were as follows: (i) age 18–30, (ii) students of physical education, (iii) healthy and not suffering from chronic diseases, (iv) non-drug users, (v) not following elimination diets (e.g., ketogenic diet), (vi) not suffering from depression (less than 13 points in Beck Depression Inventory (BDI)). Information about the study, along with an invitation to participate, was sent to all students of physical education at Jozef Pilsudski University of Physical Education in Warsaw by e-mail. All participants were recruited from April to June 2023, and they provided written consent to participate in this study. Among them, 10 did not perform all anticipated tasks or provide all necessary data, and they were excluded from the study.

### 2.2. Experimental Procedures

Body height was measured in all 172 participants with an accuracy of 0.1 cm using a stadiometer (Tanita HR-001, Tanita, Tokyo, Japan). Body weight and body composition were measured using the bioelectrical impedance (BIA) method with an accuracy of 0.1 kg using the TANITA BC-418 device (Tanita, Tokyo, Japan). BIA involves measuring the body’s electrical resistance through the flow of an electric current of a specific intensity and frequency. Prior to the measurement, the participants did not perform any physical exercise; they were at least 4 h after a meal, and during the measurement, they were dressed only in underwear. Two parallel measurements were performed for each participant. In case there was a difference between the two measurements, a third measurement was performed and an average of two similar measurements was taken. Analysis of body composition with the BIA technique allowed us to assess total body weight and body mass index (BMI) according to the following formula: body weight [kg]/height [m]^2^), fat tissue mass [kg], percentage content of fat tissue [%], fat-free mass [kg] and total body water content [%].

On the day of the study, the participants completed questionnaires concerning their diet, dietary habits, physical activity and nutritional knowledge. KomPAN^®^ Dietary Habits and Nutrition Beliefs Questionnaire for people 15–65 years old, version 1.1 (Polish version) was used [47]. The questionnaire was distributed in paper form personally to each participant, and it was filled in by the participant immediately during study in the presence of one investigator. The KomPAN^®^ questionnaire was previously validated by outstanding Polish nutritionists [48].

Venous blood samples were obtained from all of the enrolled students, collected into clot tubes and centrifuged to obtain serum samples. The separated serum samples were stored at −80 °C until FA and BDNF were analyzed. The detailed characteristics of the participants are presented in Table 1.

### 2.3. BDNF Analysis

The BDNF concentration in the serum was measured using a commercial Human BDNF ELISA kit (Thermo Fisher Scientific, Waltham, WA, USA, catalog number EH42RB). All samples were analyzed in duplicate using the manufacturer’s recommended buffers, diluents and substrates and following the manufacturer’s guidelines to ensure protocol compliance and minimize measurement errors. An eight-point standard curve was placed on each plate. Serum samples were diluted four-fold according to the manufacturer’s recommendations. The reading was performed with a microplate reader (Model Synergy H1, BioTek, Winooski, VT, USA) at a wavelength of 450 nm. The test range for BDNF was 0.066–16 ng/mL, with a sensitivity of 80 pg/mL. The intra-assay and inter-assay CV rates were <12% and <10%, respectively.

### 2.4. Fatty Acid Analysis

The serum samples were thawed only once, and three parallel 500 μL samples from one participant were subjected to hydrolysis (0.5 mL of KOH aqueous solution (4 mol/L) (KOH (Potassium hydroxide pure p. a., Pol-Aura, Morąg, Poland)). In addition, 1 mL of KOH methanolic solution (1 mol/L) (KOH (Potassium hydroxide pure p. a., Pol-Aura, Poland), methanol pure p. a., Stanlab, Lublin, Poland) was added, and the samples were flushed with Ar (argon minimum 99.999% vol., Multax, Zielonki-Parcela, Poland), heated for 15 min at 55 °C and allowed to rest at room temperature overnight. Trans-esterification to fatty acid methyl esters (FAMEs) was performed according to the previously validated procedure of Czauderna et al. [49]. Nonadecanoic acid (C19:0) (Sigma, St. Louis, MO, USA) was used as the internal standard. FAME analysis was performed using a GC-MS-QP2010 Plus EI (Shimadzu; Tokyo, Japan) equipped with a BPX70 fused silica capillary column (120 m × 0.25 mm i.d. × 0.25 μm film thickness; SHIM-POL, Warsaw, Poland), a quadrupole mass selective (MS) detector (Model 5973 N; Shimadzu; Tokyo, Japan), an injection port and helium as the carrier gas (helium analyzed pure minimum 99.999% vol., Multax, Poland). The detailed conditions of the GC/MS analyses were described previously [50]. FAME identification was based on electron impact ionization spectra of FAME and compared to authentic FAME standards (Supelco 37 Component FAME Mix, Sigma, St. Louis, MO, USA, and Bacterial Acid Methyl Ester Mix, Sigma, St. Louis, MO, USA), a c9,t11C18:2 methyl ester standard (Sigma, St. Louis, MO, USA), a t10,c12C18:2 methyl ester standard (Sigma, St. Louis, MO, USA) and the NIST 2007 reference mass spectra library (National Institute of Standard and Technology, Gaithersburg, MD, USA). All FAME analyses were based on total ion current (TIC) chromatograms and/or selected-ion monitoring (SIM) chromatograms. For quantification of the content of each assayed FA (as FAME) in assayed samples, a linear calibration equation was used [50].

### 2.5. Calculation of Indices of Serum Samples

The following indices were calculated based on the FA content of the serum samples:A-SFA = C12:0 + C14:0 + C16:0;
T-SFA = C14:0 + C16:0 + C18:0;
^A-SFA^index = (C12:0 + 4 × C14:0 + C16:0)/(ΣMUFA + Σ*n*-6PUFA + Σ*n*-3PUFA);
^T-SFA^index = (C14:0 + C16:0 + C18:0)/[(0.5 × ΣMUFA + 0.5 × Σ*n*-6PUFA + 3 × Σ*n*-3PUFA)/Σ*n*-6PUFA)];
^C18:0^∆9index = c9C18:1/(c9C18:1 + C18:0);
^Σ∆9,6,5,4^FAindex = (ΣMUFA + ΣPUFA)/(C16:0 + C18:0 + C20:0 + C22:0 + C24:0 + ΣMUFA + ΣPUFA);
*^n^*^-6ElongC20/C18^index = c11c14C20:2/(c11c14C20:2 + c9c12C18:2);
*^n^*^-3ElongC22/C20^index = c7c10c13c16c19C22:5/(c7c10c13c16c19C22:5 + c5c8c11c14c17C20:5);
^Δ4desaturation^index = c4c7c10c13c16c19C22:6/(c4c7c10c13c16c19C22:6 + c7c10c13c16c19C22:5);
^Δ5desaturation^index = c5c8c11c14C20:4/(c5c8c11c14C20:4 + c8c11c14C20:3);
h/H-Ch = (c7C18:1 + c9C18:1 + c12C18:1 + c14C18:1 + c11C20:1 + 13C22:1 + c9c12C18:2 + c9c12c15C18:3 + c6c9c12C18:3 + c5c8c11c14C20:4 + c11c14C20:2 + c5c8c11c14c17C20:5 + c7c10c13c16C22:4 + c7c10c13c16c19C22:5)/(C14:0 + C16:0);
where A-SFA is the sum of atherogenic saturated fatty acids; T-SFA is the sum of thrombogenic saturated fatty acids; ΣMUFA is the concentration sum of monounsaturated fatty acids (MUFAs); Σ*n*-6PUFA and Σ*n*-3PUFA are the concentration sums of *n*-6PUFA and *n*-3PUFA, respectively; h/H-Ch is the ratio of hypo- to hypercholesterolemic fatty acids and c—cis.

Indices were calculated on the basis of each FA determination in the serum because of their similar bioactive properties (e.g., sum of atherogenic and thrombogenic fatty acids, i.e., A-SFA and T-SFA), as well as the fact that the ultimate effect of these compounds in the organism does not simply depend on their content but is a resultant of their mutual interplay (e.g., ^ASFA^index, ^TSFA^index, h/H-Ch). Not only does the FA content in the diet influence their effects in the body, but also the endogenous metabolism of FA, e.g., by elongation and desaturation. That is why we have decided to calculate the relationships (concentration ratios) between FA products and their precursors to estimate the enzyme activities (^C18:0∆9^index, ^Σ∆9,6,5,4FA^index, *^n^*^-6ElongC20/C18^index, *^n^*^-3ElongC22/C20^index, ^Δ4desaturation^index or ^Δ5desaturation^index). We are aware that these ratios cannot be assumed to directly reflect enzyme activities, but these indices inform the final effects of enzyme actions strictly depending on their activity; similarly successful, recognized approaches were previously used as an indirect way to measure enzyme activities [51,52].

### 2.6. Statistical Analysis

For continuous variables, means and standard deviations were calculated. Categorical variables are presented as frequencies and percentages.

To verify whether food frequency consumption significantly affected group diversity, chemometric analyses were performed. Prior to the analyses, the original data were transformed into natural logarithms and then autoscaled (standardized).

Cluster analysis (CA) was performed using the agglomerative approach. Euclidean distance served as the method for distance determination, while the Ward method was applied as the agglomeration method. To analyze the dendrogram and identify clusters, the less stringent Sneath criterion (66%) was employed (Figure 1).

Moreover, to obtain appropriate classification rules for participants into distinct clusters, a linear discriminant analysis (LDA) for variables concerning the frequency of dietary product intake was performed. The relevant discriminant functions were calculated via a stepwise progressive method, with the adopted tolerance value of 1 − R^2^ = 0.01 to optimize the LDA.

Significant differences among participants classified into different clusters were established for variables meeting the criteria of a normal distribution (verified with Shapiro–Wilk test) and equality of variances (verified with Leven’s test) with one-way ANOVA with Tukey’s *post hoc* test. For FA content and values of calculated indices, significant differences were established with nonparametric tests, such as the Kruskal-Wallis test with a multiple comparison test as a *post hoc* test. For categorical variables, significant differences were established with the χ^2^ Pearson test. *p* < 0.05 was considered significant.

To assess the dependence between BDNF concentration and body mass, BMI or FA concentration Spearman correlation coefficients were calculated. They were considered significant if *p* < 0.05.

Statistica 13.3 software (TIBCO Software Inc., Palo Alto, CA, USA) was used for the statistical analysis.

## 3. Results

### 3.1. Characteristics of Participants

Descriptive analysis was performed for the whole study population (Table 1 and Table 2), which consisted solely of 172 men. The mean age of the students who participated in the study was 21.9 ± 2.6 years, the mean height was 181 ± 13 cm and the mean body weight of the participants was 81 ± 11 kg, which resulted in a mean BMI of 24.4 ± 2.9 kg/m^2^. The average fat content of the adipose tissue was 11.2 ± 4.6 kg, which corresponds to an average fat content of 13.5 ± 3.8%, whereas the mean fat-free mass was 69.6 ± 7.3 kg, from which the total body water content was 46.6 ± 5.7 kg.

Most participants (58.1%) lived in large cities (over 100,000 inhabitants), and only 5.8% of them lived in small towns and villages (5.8% and 15.7%, respectively). As the investigated population consisted mostly of students, most of the participants had attended secondary education (59.3%) or higher education (36.6%). Most of the participants worked a temporary job (46.5%) or did not work at all and declared studies to be their main occupation (33.7%). The vast majority (68.0%) of the investigated students described their financial situation as average, and only 8.7% of participants declared their financial status to be below average. The majority of participants (86.0%) reported not smoking tobacco; however, 37.8% of the investigated students smoked tobacco in the past. A total of 59.3% of the studied participants reported that their health status was better than that of others, and only 8.7% of the investigated population reported that their health status was worse than that of others. On weekdays, most of the students slept for 6–9 h per day; however, some of them had the opportunity to sleep longer on weekends (31.4% of participants slept more than 9 h per day on weekends). The investigated population consisted of more than 80% of the students of Jozef Pilsudski University of Physical Education who declared their level of physical activity to be moderate and high at work or school and during their time off (41.3% and 40.1% at work or school and 28.5% and 65.7% during their time off, respectively), which indicates that some of them undertook physical training after class. Integrated physical activity levels estimated with the KomPAN^®^ questionnaire showed that 57.0% of the investigated population had high and 30.2% had moderate levels of physical activity. Regarding their diet, the vast majority of participants described their diet as very good or good (4.7% and 70.9%, respectively), and they did not follow a special diet (86.6%). Only 13.4% of the investigated students decided to go on a diet, but it was their personal choice. Moreover, the majority of the investigated students reported no or only slight differences in their nutritional status between weekdays and weekends (48.3% and 42.4%, respectively). The vast majority of the studied population declared good or sufficient levels of nutritional knowledge (42.4% and 34.9%, respectively), and only 14.0% of them claimed to possess insufficient nutritional knowledge. The evaluation of nutritional knowledge with the KomPAN^®^ questionnaire confirmed these findings, as 15.7% of the investigated population had insufficient nutritional knowledge.

### 3.2. Diet Analysis

Completed KomPAN^®^ questionnaires obtained from each of the participants were used for analysis of food frequency consumption. The questions concerned 6 categories of increasing frequency of 33 categories of food product consumption, from “never” to “few times a day”. The detailed data describing the frequency of food consumption for the whole population are given in Appendix A.

### 3.3. Chemometric Analysis

To verify whether there were some regularities and similarities in food frequency consumption among the investigated athletes, cluster analysis (CA) was performed, and the results are presented as a dendrogram in Figure 1. The application of the less restrictive Sneath criterion (66%) to the dendrogram analysis allowed us to distinguish three clusters that differentiated the participants due to food frequency consumption (Figure 1). The first cluster (Cl1) included 23 participants, the second cluster (Cl2) included 63 participants and the remaining 86 students were included in the third cluster (Cl3).

To verify which products were of great importance for establishing dietary patterns, LDA was performed to obtain appropriate classification rules for the enrolled participants. All variables concerning the frequency of food and dietary product consumption were included in the model (part C of the KomPAN^®^ questionnaire). The relevant discriminant functions were calculated via a stepwise progressive method. In the analysis, 22 variables were included in the final LDA model, and 10 of them (sweetened hot beverages, fruit juices, vegetables, milk, wholemeal (brown) bread/bread rolls, vegetable juices or fruit and vegetable juices, butter, tinned (jar) vegetables, sweetened carbonated or still beverages, fruits) were significant in the model. All of them made comparable contributions to overall discrimination. Canonical analysis allowed us to distinguish two discriminant functions (DFs) that were statistically significant (*p* < 0.0001 for both DFs). DF1 was the most significant function, as it explained 69.33% of the discriminatory power, whereas DF2 explained 30.67% of the discriminatory power (Table 3).

Analysis of the canonical mean variables indicated that DF1 had the greatest impact on the distinction of Cl1 from Cl3 and that DF2 had the greatest impact on the distinction of Cl1 from Cl2 (Table 3). The graph analysis confirmed the results provided by the average values of the canonical variables (Figure 2).

The calculated classification matrix indicated that the average classification efficiency based on the calculated functions was 90.12% (Appendix A). For individual clusters, these coefficients were 86.96% for Cl1, 88.89% for Cl2 and 91.86% for Cl3.

### 3.4. Cluster Characteristics

The identified clusters were characterized according to the general characteristics of the participants and their dietary habits, and no differences were revealed by the χ^2^ Pearson test (*p* > 0.05) (Appendix A).

### 3.5. Diet Quality of the Clusters—Comparison of Dietary Patterns

The frequency of food consumption was analyzed for the revealed clusters using the χ^2^ Pearson test (Appendix A). Significant differences were found for various foods, including bread types, dairy, fruits, vegetables, beverages and more. Certain items, such as buckwheat, oats, wholegrain pasta, fresh cheese curd products and eggs, were excluded from the LDA model. Others, including fermented milk beverages, fish, pulse-based foods, potatoes, sweets, instant soups and energy drinks, were included but did not show statistically significant results (Table 3). The diet of Cl1 participants was characterized by frequent consumption of wholemeal bread (78.2% consumed it at least a few times a week), milk (69.5% drank it daily or more), fruits (78.2% consumed them once or several times daily), vegetables (86.9% ate them at least once daily), fruit/vegetable juices, and sweetened hot beverages (61% consumed them once or several times daily). Over 20% in Cl1 and Cl3 excluded butter, and more than 32% excluded tinned vegetables. Only 13% in Cl1, 7.9% in Cl2 and 25.6% in Cl3 eliminated sweetened carbonated/still beverages. In Cl2 and Cl3, some participants excluded wholemeal bread (15.9% and 10.5%, respectively) and dairy beverages (3–9%). In Cl3, over 45% excluded sweetened hot beverages, and 26% avoided sweetened carbonated drinks. Cl1 is marked by frequent consumption of healthy foods (vegetables, fruits, wholemeal bread) whereas Cl3 seems to be the most restrictive group, with a high percentage avoiding sweetened drinks and dairy products.

To comprehensively evaluate diet quality, three diet quality indices were calculated and interpreted—the pro-healthy diet index (pHDI), nonhealthy diet index (nHDI) and diet-quality index (DQI)—by summing the frequency of food consumption (times/day) of the indicated 10 (pHDI), 14 (nHDI) or 24 food groups (DQI) (Table 4), as described in the KomPAN^®^ questionnaire. Cl1 (pattern 1) was characterized by significantly greater mean values of pHDI (*p* < 0.0001) and DQI (*p* < 0.0001) than Cl2 (pattern 2) and Cl3 (pattern 3), which indicates a higher quality diet and a greater intensity of beneficial dietary characteristics for health. Cl2 was characterized by a significantly greater nHDI than Cl3 (*p* = 0.0192), which indicates a worse quality of diet and a greater intensity of dietary characteristics that are harmful to human health. These observations were further confirmed by the frequency of interpretation of established indices in the revealed clusters. Cl1 was characterized by a greater frequency of medium pHDI values and a greater frequency of high-intensity unhealthy diet characteristics, whereas Cl2 and Cl3 were characterized by a greater frequency of low pHDI values (*p* < 0.0001 and *p* = 0.0055, respectively) (Table 4).

### 3.6. Fatty Acid Profile of Clusters

The overall profile of FA in the serum samples is given in Table 5. In the present study, 52 fatty acids were identified, including 17 saturated fatty acids (SFAs) with 6 odd and branched fatty acids (OBCFAs) and 19 monounsaturated fatty acids (MUFAs), including 2 trans isomers and 16 polyunsaturated fatty acids (PUFAs) with 3 conjugated linoleic acid (CLA) isomers. The total content of SFA, MUFA and PUFA were comparable. Palmitic (C16:0), oleic (c9C18:1) and linoleic (c9c12C18:2) acids were the main FAs present in the serum samples. A comparison of the FA content of the serum samples of participants belonging to three different clusters revealed that 8 SFAs were significantly more abundant in Cl1 than in Cl3, resulting in a significantly greater total SFA content in Cl1 than in Cl3. The contents of 11 MUFAs differed significantly among the revealed clusters; the amounts of c9C14:1, c9C16:1, c10C17:1, c11C18:1 and c11C20:1 were significantly lower in Cl3 than in Cl1 and Cl2, and the amounts of other differing MUFAs were significantly lower in Cl3 than in Cl1. In the case of PUFA, the contents of 7 FAs significantly differed among the clusters. Cl1 was characterized by significantly greater contents of c11c14c17C20:3, c10c13c16c19C22:4 and c7c10c13c16c19C22:5 than Cl3, whereas the Cl2 contents of c9t11C18:2 and c11c14C20:2 significantly exceeded their content in Cl3. The amounts of c6c9c12C18:3 and c8c11c14C20:3 in Cl3 were significantly lower than those in the other two clusters (Cl1 and Cl2).

For atherogenic SFA (A-SFA) and thrombogenic SFA (T-SFA), the highest levels were detected in Cl1 (significantly greater than those in Cl3). Similarly, the ^C18:0∆9^index was significantly greater in CI1 than in Cl3. The opposite difference was detected for the ^∆5desaturation^index, for which the value in Cl3 was the highest and significantly exceeded the value calculated for Cl1. In terms of the other indices, no significant differences between the revealed clusters were found (Appendix A).

### 3.7. BDNF Content

The mean concentration of BDNF in the serum of the investigated individuals was 9.1 ± 3.6 ng/mL, ranging from 0.8 ng/mL to 19.1 ng/mL. The amount of BDNF was negatively correlated with body mass (R = −0.2159, *p* = 0.0047; Figure 3A) and BMI (R = −0.1757, *p* = 0.0219; Figure 3B). There were no other correlations between BDNF concentration and FA concentration, physical activity level or diet quality. We compared the BDNF content among the revealed clusters, and no significant differences were observed (*p* = 0.1275, Figure 4A). Similarly, there were no significant differences in BDNF concentration among groups of individuals with different levels of physical activity (*p* = 0.1309, Figure 4B) or with different levels of nutritional knowledge (*p* = 0.6900, Figure 4C). Within the revealed clusters, no additional correlations between BDNF concentration and FA content were detected.

## 4. Discussion

According to the data regarding the physical activity of Polish citizens in 2023, only 28% of them regularly performed different forms of physical exercise (excluding walking) and met the WHO recommendation [53]. On the other hand, numerous studies have confirmed the beneficial influence of physical activity on body condition and overall functioning, as well as the strong connection between physical and mental abilities in elderly individuals, adults and children (e.g., in children with dyslexia [2]), in older adults with mild cognitive impairment [54] or in adult patients with depression [55].

Among the different plausible mechanisms of this phenomenon, its impact on BDNF levels has been widely investigated. Research on animals has shown an increase in BDNF levels following physical activity in several brain regions, such as the hippocampus, prefrontal cortex, motor cortex, lateral septum, cerebellum, striatum and amygdala [37]. Numerous studies as well as meta-analyses have confirmed the influence of physical activity on BDNF levels and improvements in mood and cognitive functions in humans [8,35,37,39,56,57,58]. Both mRNA BDNF and BDNF protein expression levels in skeletal muscle have been shown to increase in response to muscle contraction, which is associated with enhanced lipid oxidation [59]. Pereira et al. revealed that strength exercise significantly increased the plasma levels of BDNF, whereas no differences were found after aerobic exercise. They claim that chronic elevation in plasma BDNF levels after strength exercise may be partially related to an increase in the production of this neurotrophin in the muscles. Strength exercise involves large volumes of muscle mass, which does not occur in aerobic exercise [57]. Cefis et al. observed that the intensity and modality of exercise have a gradual effect on the BDNF pathway in the hippocampus. Only the highest intensity leads to the activation of this pathway in the prefrontal cortex, which indicates improvement of memory performance after the most intense exercise [39].

However, some studies of athletes have shown slight differences. The baseline serum level of BDNF was significantly lower in the athlete group compared to the control group with a sedentary lifestyle [28,29]. Hence, a significant negative correlation between the serum BDNF concentration and total energy expenditure (r = −0.507, *p* < 0.05), movement-related energy expenditure (r = −0.503, *p* < 0.05) and walking count (r = −0.480, *p* < 0.05) was observed. The authors suggest that vigorous habitual physical activity decreases the serum BDNF concentration [29]. This finding contradicts the results of other studies, in which higher BDNF concentrations in the blood were observed after physical exercise, both in athletes and in sedentary subjects [7,28,29,30,31,32,33,34]. Suwa et al., who also observed increased serum BDNF levels both in active and sedentary subjects, demonstrated that during the recovery phase, increased serum BDNF decreases below the baseline level in active subjects. In the sedentary group, the observed decrease was less pronounced. The authors of the mentioned study suggested that one of the possible physiological roles of BDNF is the repair of exercise-induced muscle damage. As a physical activity effect in muscle damage, active participants may have adapted to utilize circulating BDNF for the promotion of muscle repair, which results in more pronounced decrease in BDNF level in comparison with sedentary participants [59]. The participants of our study were young adult male students from the Faculty of Physical Education at Jozef Pilsudski University of Physical Education in Warsaw who declared their level of physical activity to be moderate or high. All of them took up physical exercise regularly, which may be the reason for lack of significant differences in BDNF concentrations among subgroups differing in declared level of physical activity. However, a tendency toward a decrease in BDNF concentration with an increase in physical activity level was observed (Figure 4C). These results are in accordance with the previously mentioned observations made in physically active subjects, as higher physical activity of the investigated population was connected with higher muscle damage and, as a consequence, more effective utilization of BDNF in muscle repair. The timepoint of sampling seems to be crucial for drawing conclusions about the impact of physical activity on BDNF level. We presume that in recovery, stationary phase BDNF levels seem to be lower in physically active individuals than in sedentary ones, whereas in the physical activity phase, BDNF levels increase in both groups proportionally to the intensity of the activity. In our study, serum samples from all participants were taken in a stationary recovery phase, in which the tendency to decreased BDNF concentration in relation to physical activity level was observed. It seems of utmost importance to investigate the kinetics of BDNF formation and utilization during physical exercises and in the recovery phase, as well comparing different types of exercise and different length of training.

BDNF can influence metabolic processes such as energy balance, appetite control, glucose metabolism and insulin sensitivity [25]. Moreover, BDNF plays a crucial role in controlling body weight and energy homeostasis. It was observed that increased levels of BDNF reduced food consumption and maintained energy balance, and TrkB activation by BDNF was essential for appetite regulation and energy homeostasis [24]. In the present study, negative correlations of BDNF concentration with body mass (R = −0.2159, *p* = 0.0047) and subsequently with BMI (R = −0.1757, *p* = 0.0219) were observed, which confirms the interplay between BDNF level and energy balance. Additionally, Taha et al. revealed that BDNF levels were significantly lower in people with obesity classes II and III than in those with a normal weight (*p* < 0.05), and the Spearman rank correlation test revealed a statistically significant negative correlation between BMI and BDNF (R = −0.478, *p* < 0.01) [22]. Another previous study suggested that BDNF plays an essential role in energy metabolism by altering skeletal muscle fat oxidation in an adenosine monophosphate-activated protein kinase-dependent manner [57].

Our previous studies confirmed that different dietary factors influence the FA profile in serum [60,61,62]. On the other hand, many studies suggest that specific dietary changes could be an effective way to exert an impact on BDNF levels. These premises allowed us to verify whether there are specific dietary patterns in the investigated athletes, which allowed us to evaluate the quality of their diet and which factors are responsible for such differences, as well as whether there is a strict connection between diet quality and BDNF and FA levels. We also checked whether there were any dependencies between BDNF and FA content in the serum. The health-related quality of a diet, particularly in terms of adhering to quantitative and qualitative dietary guidelines, can be evaluated using several indices, such as the pro-healthy diet index (pHDI), non-healthy diet index (nHDI) and the overall diet quality index (DQI), which are based on the frequency of food consumption (times/day) of the indicated 10 (pHDI), 14 (nHDI) or 24 food groups (DQI). This approach has been widely used not only by Polish scientists using the KomPAN^®^ questionnaire [63,64,65,66,67] but also by other scientists who created or applied similar diet quality indices [68,69].

A rational diet is of great importance for health and achieving satisfactory sports performance. However, the optimal diet for achieving peak performance varies from athlete to athlete, and attempts are made to determine the personalized diet that will be the best for a particular sportsman. Regarding the diet of athletes, most studies have focused on nutrient intake rather than eating patterns [70]. Dietary patterns focus on the impact of the whole diet rather than examining individual nutrients and products within that diet and provide a stronger format for understanding the complex relationship between diet and health [71]. In the present study, dietary patterns were empirically extracted using cluster analysis (CA) and linear discriminant analysis (LDA), which are multivariate statistical procedures. This chemometric approach allowed us to verify whether there are some regularities and similarities in food frequency consumption among the investigated physically active men (using the CA) and to identify the dietary factors responsible for the determining of those dietary patterns (using the LDA). A similar approach, involving the use of multivariate statistical procedures to identify dietary patterns in athletes, was also used by other authors [71,72]. In the analysis, 22 variables were included in the final LDA model, and 10 of them (sweetened hot beverages, fruit juices, vegetables, milk, wholemeal (brown) bread/bread rolls, vegetable juices or fruit and vegetable juices, butter, tinned (jar) vegetables, sweetened carbonated or still beverages, fruits) were significant in the model. All of them made comparable contributions to overall discrimination. The relevant discriminant functions were calculated via a stepwise progressive method, which allowed us to distinguish all three revealed clusters. Cl1 contained only 23 participants, but its diet was evaluated as having the greatest impact on the mean values of pHDI (*p* < 0.0001) and DQI (*p* < 0.0001). Participants in cluster Cl1 consumed wholemeal bread more frequently, drank milk more often, ate fruits and vegetables more regularly, consumed more fruit/vegetable juices, and drank sweetened hot beverages frequently. On the other hand, the diet of participants included in Cl2 (n = 63) was evaluated to be of the worst quality, as indicated by the higher nHDI values. Although no significant differences were detected in BDNF concentration among the revealed clusters (*p* = 0.2175), a possible trend was observed, with the highest BDNF levels observed for Cl1 (Figure 4A), which is the cluster with the best diet quality (9.9 ± 3.6 ng/mL in Cl1 vs. 9.4 ± 3.9 ng/mL in Cl2 vs. 8.6 ± 3.3 ng/mL in Cl3, respectively). This observation seems to partially confirm the impact of proper diet on BDNF content in serum. Regarding the FA concentration in serum and the values of indices calculated based on the FA profile, only several differences (4 out of 26) were observed among the revealed clusters, which indicates a small impact of dietary patterns on the FA content in the serum of physically active men (Appendix A). Moreover, no correlations between BDNF and FA, especially *n*-3PUFA, were detected in the present study. Many authors and numerous meta-analyses emphasize the positive impact of *n*-3PUFA supplementation on BDNF levels [5,23,25,73,74,75,76]. The fact that no such dependence was observed in the present study may be because no additional supplementation of *n*-3PUFA was applied in this study, and participants occasionally consumed fish and seafood (which are the richest dietary sources of *n*-3PUFAs).

In the present study, BDNF and FA concentrations were determined in serum samples, where the concentration of this neurothrophin was more than 50-fold greater than that in plasma [8]. In the periphery, platelets store BDNF, and these cells are considered the major reservoir of circulating BDNF, as more than 90% of blood BDNF is stored in platelets. Once activated, platelets release BDNF during the clotting process. This process is considered the main mechanism to explain differences between serum and plasma concentrations [8]. The serum BDNF concentration seems to reflect both platelet-stored BDNF and freely circulating BDNF in the blood, while the plasma BDNF concentration seems to reflect only freely circulating BDNF [59].

We decided to choose the physically active male population to examine how different dietary factors may influence BDNF levels. As diet composition is one of the crucial factors influencing sport performance, it was expected that physical education students possess proper nutritional knowledge. The vast majority of the studied population possessed good or sufficient levels of nutritional knowledge (17.4% and 66.9%, respectively), and only 15.7% of the investigated population reported insufficient nutritional knowledge. It was expected that proper nutritional knowledge would exert a positive effect on BDNF levels. Notably, the BDNF concentration tended to positively correlate with the level of nutritional knowledge (Figure 4C). Comparing the level of nutritional knowledge with revealed nutritional patterns, it is noteworthy to mention that numerosity of participants with good nutritional knowledge (n = 30) and participants included in Cl1 (n = 23), which had the best diet quality, was similar. However, nutritional knowledge was not reflected by the best quality of diet in every student.

The present study is not without limitations. It was limited to only one gender, which restricts the generalizability of the results. Additionally, the research focused solely on physically active men, with no comparisons made between physically active and non-active men; nor were different forms, intensities or durations of physical exercise studied. In order to obtain comprehensive data, future research should be conducted among a larger number of participants of different genders and should address the effects of different durations and intensities of physical activity, as well as different exercise types, on BDNF and lipidomic profiles in long-term studies. However, the investigation of the kinetics of BDNF formation and utilization in active and recovery phases seems to be of utmost importance in the future. Regarding the limitations of the ELISA method, which is considered to be a semi-quantitative method, all measurements were performed in duplicate. Regarding fatty acid determination in serum with the GC-MS technique, to decrease random errors and increase the precision of measurement, three parallel samples of serum from one participant were investigated.

## 5. Conclusions

The research objective was achieved by conducting detailed analyses that allowed us to identify significant correlations between diet quality and BDNF and FA levels. This study fills an interesting research gap, although some sections require further improvement. Based on the results of our study, we confirmed the hypothesis that diet quality affects BDNF content; however, such a confirmation was not achieved for FA concentrations in the serum of physically active individuals. Our studies revealed that there is a negative correlation between the amount of BDNF and body mass and BMI in physically active men. BDNF concentration tends to negatively correlate with the level of physical activity and positively correlate with the nutritional knowledge of participants. Three revealed dietary patterns differed in frequency of food consumption, which interspersed with diet quality, but no correlations between BDNF concentrations and FA contents were detected. Our studies provide valuable insights into dependencies among physical activity, quality of diet and BDNF level. We presume that they depend on each other, correlating and interacting to provide optimal health, physical performance, energy balance, cognitive functions and mental status. Further research, especially long-term studies, are needed among a larger number of participants of different genders, addressing the effects of different durations and intensities of physical activity, as well as different exercise types, on BDNF and lipidomic profile. However, the investigation of the kinetics of BDNF formation and utilization in active and recovery phases seems to be of utmost importance in the future.

## Figures and Tables

**Figure 1 nutrients-16-04326-f001:**
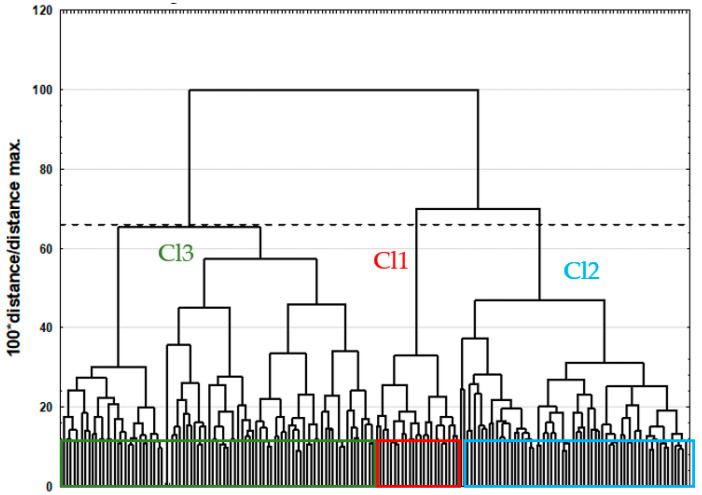
Dendrogram of diet similarities of participants. The application of the less restrictive Sneath criterion (66%) allowed us to distinguish three clusters that differentiated the participants due to food frequency consumption.

**Figure 2 nutrients-16-04326-f002:**
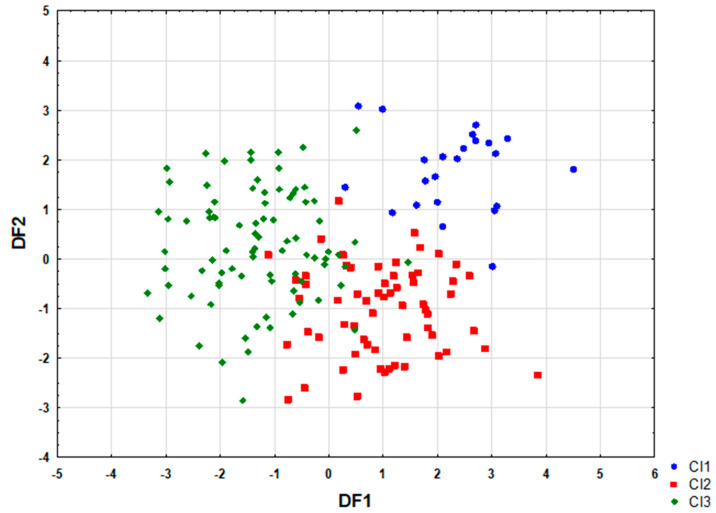
Scatterplot of canonical values for functions DF1 and DF2. DF—discriminant function, Cl—cluster.

**Figure 3 nutrients-16-04326-f003:**
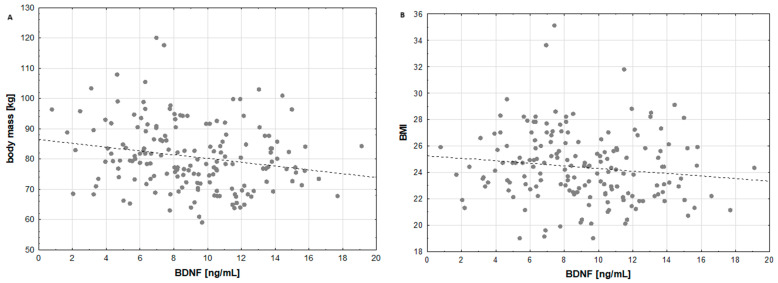
Correlation between brain-derived neurotrophic factor (BDNF) concentration and body mass (**A**) and correlation between brain-derived neurotrophic factor (BDNF) concentration and body mass index (BMI) (**B**). BDNF—brain-derived neurotrophic factor; BMI—body mass index.

**Figure 4 nutrients-16-04326-f004:**
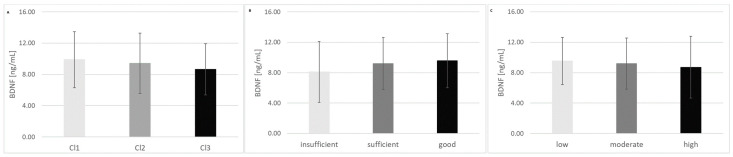
Serum brain-derived neurotrophic factor (BDNF) concentrations in the revealed clusters (**A**), serum brain-derived neurotrophic factor (BDNF) concentrations in participants with different levels of physical activity (**B**) and concentration of brain-derived neurotrophic factor (BDNF) in the serum of participants with diverse nutritional knowledge (**C**). BDNF—brain-derived neurotrophic factor; Cl—cluster.

**Table 1 nutrients-16-04326-t001:** Characteristic of the participants.

age [years]	21.9 ± 2.6
height [cm]	181 ± 13
body mass [kg]	81 ± 11
BMI	24.4 ± 2.6
fat [%]	13.5 ± 3.8
fat mass [kg]	11.2 ± 4.6
fat-free mass [kg]	69.6 ± 7.3
total body water [kg]	46.6 ± 5.7

The continuous variables were presented as means ± standard deviation (SD), BMI—body mass index.

**Table 2 nutrients-16-04326-t002:** Characteristics of the participants based on the KomPAN^®^ Questionnaire.

Personal Diet	Answer	n	[%]
sex	Male	172	100.0
	Female	0	0.0
place of residence	Village	27	15.7
	Town below 20,000 inhabitants	10	5.8
	Town between 20,000 and 100,000 inhabitants	35	20.3
	City over 100,000 inhabitants	100	58.1
financial situation	Below average	15	8.7
	Average	117	68.0
	Above average	40	23.3
work status	No, I am retired or receiving a disability living allowance	2	1.2
	No, I am on maternity leave, I am unemployed or other (housewife/househusband)	0	0.0
	Yes, but it is only a temporary job	80	46.5
	Yes, I am permanently employed	32	18.6
	No, I study	58	33.7
education	Primary	5	2.9
	Lower secondary	2	1.2
	Upper secondary	102	59.3
	Higher (e.g., BSc, MSc)	63	36.6
following a diet	No	149	86.6
	Yes, as advised by my doctor for medical reasons	0	0.0
	Yes, it was my personal decision	23	13.4
physical activity at work or school	Low	32	18.6
	Moderate	71	41.3
	High	69	40.1
physical activity during time off	Low	10	5.8
	Moderate	49	28.5
	High	113	65.7
integrated physical activity level	Low	22	12.8
	Moderate	52	30.2
	High	98	57.0
tobacco smoking	No	148	86.0
	Yes	24	14.0
tobacco smoking in the past	No	107	62.2
	Yes	65	37.8
hours of sleep (weekdays)	≤6 h/day	54	31.4
	6–9 h/day	114	66.3
	>9 h/day	4	2.3
hours of sleep (weekends)	≤6 h/day	10	5.8
	6–9 h/day	108	62.8
	>9 h/day	54	31.4
health status	Worse than others	15	8.7
	The same as others	55	32.0
	Better than others	102	59.3
nutrition knowledge (self-evaluation)	Insufficient	24	14.0
	Sufficient	60	34.9
	Good	73	42.4
	Very good	15	8.7
nutrition knowledge (actual)	Insufficient	27	15.7
	Sufficient	117	68.0
	Good	28	16.3
diet (self-evaluation)	Very bad	4	2.3
	Bad	38	22.1
	Good	122	70.9
	Very good	8	4.7
weekdays to weekends diet comparison	No difference really	83	48.3
	Differs slightly	73	42.4
	Very different	16	9.3

The categorical variables were presented using frequencies (n) and percentages.

**Table 3 nutrients-16-04326-t003:** Coefficients and average values of canonical variables included in the final model.

	Coefficients of Canonical Variables	
	DF1	DF2
Variables	69.33%	30.67%
sweetened hot beverages	0.45698	−0.07314
fruit juices	0.31380	−0.21951
vegetables	−0.00431	0.42024
milk	0.18186	0.36273
wholemeal (brown) bread/bread rolls	−0.09514	0.40305
vegetable juices or fruit and vegetable juices	0.41244	0.18746
butter	0.35040	−0.09224
tinned (jar) vegetables	−0.23662	0.18254
sweetened carbonated or still beverages	0.24726	0.09079
fruit	0.33944	0.12244
sweets	0.19758	0.05536
potatoes	−0.16290	−0.30576
energy drinks	0.17769	0.10012
white bread and bakery products	−0.18347	−0.13335
white rice, white pasta, fine-ground groats	0.22413	0.04922
fish	0.22278	0.04129
fermented milk beverages	−0.05566	0.18253
white meat	−0.04400	−0.33288
pulse-based foods	−0.05754	0.24726
instant soups or ready-made soups	0.24594	−0.10873
fried foods	−0.11565	0.28788
fast foods	0.03478	−0.25107
	**Average values of canonical variables**	
Cl1	2.26590	1.78728
Cl2	0.97553	−1.01918
Cl3	−1.32063	0.26862

DF—discriminant function; Cl—cluster.

**Table 4 nutrients-16-04326-t004:** Evaluation of the quality of diet with diet quality indices.

		Cl1		Cl2		Cl3		*p* Value
pHDI		35.6 ± 9.0 ^a,b^		16.7 ± 6.6 ^a,c^		22.5 ± 10.4 ^b,c^		<0.0001
nHDI		26.1 ± 9.1		27.2 ± 10.2 ^a^		22.7 ± 10.0 ^a^		0.0192
DQI		17.0 ± 10.3 ^a,b^		−2.7 ± 10.0 ^a,c^		6.3 ± 13.4 ^b,c^		<0.0001
		n	%	n	%	n	%	
pHDI	Low	8	34.78%	62	98.41%	76	88.37%	<0.0001
	Medium	15	65.22%	1	1.59%	9	10.47%	
	High	0	0.00%	0	0.00%	1	1.16%	
nHDI	Low	18	78.26%	48	76.19%	74	86.05%	0.2858
	Medium	5	21.74%	15	23.81%	12	13.95%	
	High	0	0.00%	0	0.00%	0	0.00%	
DQI	High intensity of non-healthy dietary characteristics	0	0.00%	3	4.76%	0	0.00%	0.0055
	Low intensity of non-healthy and pro-healthy dietary characteristics	19	82.61%	60	95.24%	80	93.02%	
	High intensity of pro-healthy dietary characteristics	4	17.39%	0	0.00%	6	6.98%	

Data are shown as mean values ± standard deviation (SD). *p*-value ≤ 0.05—significant differences among groups in one-way ANOVA. Values sharing a letter in one row are significantly different (*p* < 0.05) in the RIR Tukey test. pHDI—pro-healthy diet index, nHDI—non- healthy diet index, DQI—diet-quality index, Cl—cluster.

**Table 5 nutrients-16-04326-t005:** Content of fatty acids in serum of individuals of revealed clusters.

Fatty Acids [ug/mL]	Cl1	Cl2	Cl3	*p* Value
C8:0	1.2 ± 1.2	1.15 ± 0.95	0.82 ± 0.73	n.s.
C9:0	1.7 ± 1.5	1.4 ± 1.3	1.14 ± 0.94	n.s.
C10:0	1.8 ± 1.8	1.8 ± 1.9	1.3 ± 1.5	n.s.
C12:0	3.7 ± 3.1	3.1 ± 3.8	2.5 ± 2.6	n.s.
C14:0	18 ± 15 ^a^	15 ± 17	10.3 ± 7.3 ^a^	0.0112
i-C15:0	0.73 ± 0.25	0.78 ± 0.79	0.57 ± 0.52	n.s.
a-C15:0	1.3 ± 1.4	1.1 ± 1.0	0.80 ± 0.56	n.s.
C15:0	3.8 ± 3.8 ^a^	3.2 ± 2.9	2.2 ± 1.5 ^a^	0.0334
i-C16:0	2.5 ± 2.1 ^a^	2.3 ± 2.1	1.6 ± 0.9 ^a^	0.0158
C16:0	288 ± 232 ^a^	267 ± 251	183 ± 130 ^a^	0.0142
i-C17:0	1.4 ± 1.3 ^a^	1.2 ± 1.3	0.81 ± 0.60 ^a^	0.0098
a-C16:0	1.7 ± 1.7	1.6 ± 1.4	1.08 ± 0.85	n.s.
3,7,11,15-methylC16:0	0.36 ± 0.01	0.34 ± 0.10	0.41 ± 0.13	n.s.
C17:0	4.0 ± 3.4 ^a^	3.9 ± 3.5	2.6 ± 1.5 ^a^	0.0415
C18:0	82 ± 66 ^a^	84 ± 80	57 ± 43 ^a^	0.0417
C20:0	1.7 ± 1.9 ^a^	1.4 ± 1.4	0.94 ± 0.68 ^a^	0.0068
C22:0	1.2 ± 1.2	1.1 ± 1.0	0.81 ± 0.42	n.s.
SFA	411 ± 331 ^a^	387 ± 358	266 ± 187 ^a^	0.0156
c7C14:1	1.5 ± 1.3	1.4 ± 1.4	0.96 ± 0.68	n.s.
c9C14:1	2.5 ± 2.3 ^a^	2.0 ± 1.8 ^b^	1.30 ± 0.96 ^a,b^	0.0006
c10C15:1	1.6 ± 1.6	1.3 ± 1.1	0.89 ± 0.46	n.s.
t9C16:1	1.3 ± 1.2	0.72 ± 0.36	0.94 ± 0.62	n.s.
c7C16:1	6.1 ± 4.7 ^a^	5.6 ± 5.2	4.1 ± 3.4 ^a^	0.0311
c9C16:1	23 ± 18 ^a^	18.1 ± 18.0 ^b^	11.5 ± 8.9 ^a,b^	0.0005
c11C16:1	1.6 ± 1.3	1.6 ± 1.4 ^a^	1.02 ± 0.79 ^a^	0.0026
c9C17:1	0.73 ± 0.40	0.64 ± 0.46	0.67 ± 0.51	n.s.
c10C17:1	2.2 ± 2.1 ^a^	1.9 ± 2.0 ^b^	1.2 ± 1.0 ^a,b^	0.0065
t11C18:1	2.0 ± 1.6	1.7 ± 1.3	2.3 ± 7.1	n.s.
c6C18:1	0.53 ± 0.32	0.52 ± 0.33	0.54 ± 0.45	n.s.
c7C18:1	1.3 ± 1.3	1.2 ± 1.1 ^a^	1.0 ± 1.5 ^a^	0.0415
c8C18:1	1.2 ± 1.1 ^a^	0.94 ± 0.82	0.68 ± 0.57 ^a^	0.0024
c9C18:1	260 ± 185 ^a^	262 ± 280	172 ± 154 ^a^	0.0065
c10C18:1	2.0 ± 1.9	7 ± 23	5 ± 23	n.s.
c11C18:1	16 ± 12 ^a^	15 ± 14 ^b^	10.2 ± 7.7 ^a,b^	0.0039
c12C18:1	1.3 ± 1.3 ^a^	1.3 ± 1.5	0.79 ± 0.52 ^a^	0.0415
c14C18:1	0.98 ± 0.92	0.78 ± 0.75	0.72 ± 0.48	n.s.
c11C20:1	2.4 ± 1.8 ^a^	2.3 ± 2.0 ^b^	1.6 ± 1.3 ^a,b^	0.0359
MUFA	326 ± 233 ^a^	323 ± 325 ^b^	215 ± 179 ^a,b^	0.0052
t9t12C18:2	1.3 ± 1.1	1.18 ± 0.96	1.20 ± 0.98	n.s.
c9c12C18:2	389 ± 348	369 ± 334	264 ± 187	n.s.
c6c9c12C18:3	4.2 ± 3.4 ^a^	3.9 ± 3.6 ^b^	2.7 ± 2.5 ^a,b^	0.0032
c9c12c15C18:3	6.9 ± 6.2	6.7 ± 6.9	5.3 ± 7.5	n.s.
c9t11C18:2	1.5 ± 1.4	1.5 ± 1.2 ^a^	1.04 ± 0.69 ^a^	0.0311
c11t13C18:2	1.6 ± 1.5	1.19 ± 0.87	0.97 ± 0.50	n.s.
t9t11C18:2	1.02 ± 0.57	0.96 ± 0.49	0.88 ± 0.29	n.s.
c11c14C20:2	1.7 ± 1.5	1.7 ± 1.4 ^a^	1.08 ± 0.77 ^a^	0.0046
c8c11c14C20:3	1.7 ± 1.4 ^a^	1.6 ± 1.6 ^b^	1.03 ± 0.75 ^a,b^	0.0084
c11c14c17C20:3	10.9 ± 7.6 ^a^	11 ± 10	7.3 ± 5.4 ^a^	0.0030
c5c8c11c14C20:4	64 ± 70	62 ± 54	49 ± 35	n.s.
c5c8c11c14c17C20:5	5.3 ± 5.7	3.9 ± 3.5	3.4 ± 3.0	n.s.
c7c10c13c16C22:4	2.0 ± 2.2	1.8 ± 1.6	1.4 ± 1.2	n.s.
c10c13c16c19C22:4	1.14 ± 0.91 ^a^	1.06 ± 0.91	0.74 ± 0.42 ^a^	0.0272
c7c10c13c16c19C22:5	3.1 ± 2.5 ^a^	3.0 ± 2.6	2.1 ± 1.6 ^a^	0.0327
c4c7c10c13c16c19C22:6	11 ± 11	9.0 ± 7.5	6.6 ± 3.8	n.s.
PUFA	503 ± 454	477 ± 421	346 ± 238	n.s.
*n*-3PUFA	38 ± 32	34 ± 29	25 ± 17	n.s.
*n*-6PUFA	463 ± 422	441 ± 393	320 ± 222	n.s.
FA total	1242 ± 998 ^a^	1189 ± 1086	828 ± 597 ^a^	0.0171

Data are shown as mean values ± standard deviation (SD). *p*-value ≤ 0.05—significant differences among clusters in Kruskal–Wallis test. Values sharing a letter in one row are significantly different (*p* < 0.05) in multiple-comparisons test. Cl—cluster, SFAs—saturated fatty acids, MUFAs—monounsaturated fatty acids, PUFAs—polyunsaturated fatty acids, FAs—fatty acids, c—cis, t—trans, i—iso, a—anteiso, n.s.—not significant, *p* > 0.05 in Kruskal–Wallis test.

## Data Availability

All data are available in the main text and in the Supplemental Digital Content. Additional data related to this study will be made available by the corresponding author upon reasonable request.

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
