# Peer review of "Dietary Patterns, Serum BDNF and Fatty Acid Profiles in Physically Active Male Young Adults: A Cluster Analysis Study"

_nutrients, 2024, doi:10.3390/nu16244326_

Round 1

Reviewer 1 Report

Comments and Suggestions for Authors

Thank you for the opportunity to review this manuscript. I found the topic interesting, I like the authors' idea, but I have some comments. Firstly, only one group is included. Another thing, the authors discussed in the introduction about the effect of BDNF on the central nervous system. Still, in the present manuscript, the authors didn't assess the mental health of the participants. The authors suggest that the students are athletes, but the definition of an athlete is different. In the present manuscript, the authors couldn't control the diet of the groups. The idea of the manuscript is very good, but the authors didn't assess other markers and didn't correlate those markers with physical performance for example. 

Reviewer 2 Report

Comments and Suggestions for Authors

General comments

This study aimed to evaluate diet quality, identify specific dietary patterns, and assess their influence on BDNF and FA levels in serum. This is a study with a very interesting research gap; however, some sections require improvements.

Specific comments

Introduction:

- You should add some evidence about previous studies on Dietary Patterns, Serum BDNF, and Fatty Acid Profiles in Physically Active Young Adults.You should also clarify whether there are other studies on cluster analysis already published.

- After presenting the objectives of the study, you should add the study hypotheses. 

Material and Methods:

- The subchapter ‘2.1 Characteristics of participants’ should be divided into 2/3 to explain the representativeness of the sample (1), the study design (2) and the procedures. you should also make it clear what the procedures are and their validity, and then define the variables and their units of measurement. 

- The formulae in ‘2.4. Calculation of indices of serum samples’ should be presented more cleanly. Also, the cut-off values for qualitative analysis are missing from all the subchapters.

Results: 

- Table 1 should be split into two. One for numerical data, the other for categorical data. 

- In table 2, the cluster analysis significances should be reported. 

- Why are tables 3 and 4 presented after the cluster analysis? Wouldn't it make sense first, given that it's a comparative analysis?

- The regression in figure 3 should be contextualised in the methodology (section 2.5).

Discussions and conclusions: the first paragraph should confirm or reject the study hypotheses and check that the study objectives have been met. The rest of the discussion is extensive and very well supported, as are the conclusions. 

Comments on the Quality of English Language

Nothing to report.

Reviewer 3 Report

Comments and Suggestions for Authors

The study is exciting but needs to be clarified. It requires many points to be answered.

1. The abstract must present the problem clearly with a hypothesis for the study.

2. The introduction is very long and focuses little on the study. I suggest it be simplified and more targeted at the problem. I want to ask specific questions about the survey.

3. The methods need to be more precise and better presented. Many equations are mixed, making it impossible to know where one ends and the other begins. The entire methodology must be rewritten clearly and well formatted to be reproduced if necessary.

4. The results could be better quality; Table 1 is long and does not have formatting. Figure 1 needs to understand something that is intended to be conveyed. All tables and graphs must be reviewed, as they do not allow any conclusions to be drawn. They are often just data descriptions with little value for advancing the field.

5. The discussion needs to be better presented and more focused on the problem of the study presented. In addition to being long, it requires a clear focus on discussing the results.

Round 2

Reviewer 1 Report

Comments and Suggestions for Authors

The authors responded to all my questions, therefore I agree with the publication of the manuscript. 

Reviewer 3 Report

Comments and Suggestions for Authors

Dear authors,

Thanks for your reply. My questions were answered. The new version is better. 

Congrats on the excellent work.